**M E T H O D**                                                          **Open Access**

# Chronos: a cell population dynamics model of CRISPR experiments that improves inference of gene fitness effects

Joshua M. Dempster[1] , Isabella Boyle[1], Francisca Vazquez[1], David E. Root[1], Jesse S. Boehm[1], William C. Hahn[1,2], Aviad Tsherniak[1] and James M. McFarland[1*]

* Correspondence: jmmcfarl@
broadinstitute.org
[1]Broad Institute of MIT and Harvard,
415 Main Street, Cambridge, MA
02142, USA
Full list of author information is
available at the end of the article

## Abstract

CRISPR loss of function screens are powerful tools to interrogate biology but exhibit a number of biases and artifacts that can confound the results. Here, we introduce Chronos, an algorithm for inferring gene knockout fitness effects based on an explicit model of cell proliferation dynamics after CRISPR gene knockout. We test Chronos on two pan-cancer CRISPR datasets and one longitudinal CRISPR screen. Chronos generally outperforms competitors in separation of controls and strength of biomarker associations, particularly when longitudinal data is available. Additionally, Chronos exhibits the lowest copy number and screen quality bias of evaluated methods. Chronos is available at https://github.com/broadinstitute/chronos.

## Background

Genome-wide and large sub-genome loss of function CRISPR screens are increasingly important tools for understanding gene function in both normal and disease states. In a typical experiment, cells are infected with a library of single-guide RNAs (sgRNAs) targeting genes of interest. The CRISPR-Cas9 system is less prone to the widespread off-target effects that occur in RNAi experiments [1]. However, a number of other artifacts have been observed in pooled CRISPR screens which can complicate our ability to identify the true effect of gene knockout on cell fitness. These challenges include how to interpret discrepant data for sgRNAs targeting the same gene, including identifying and correcting for variable sgRNA efficacy [2]; correct for nonspecific CRISPR-cutting induced toxicity, which causes a gene-independent depletion of sgRNAs targeting amplified regions [3]; reduce bias when comparing screens due to variable screen quality [4]; and address incomplete phenotypic penetrance due to heterogeneity in double-stranded break repair outcomes [5].

A number of methods have been developed to address various combinations of these concerns. To combine sgRNA results into gene scores in a more robust manner than naive averaging, RIGER [6], RSA [7], and STARS [8] use statistical tests of guide rank

significance to generate gene scores, while screenBEAM uses a Bayesian hierarchical model where variation across reagents are modeled as random effects [9]. The Bayesian Analysis of Gene Essentiality (BAGEL [10]) and BAGEL2 [11] algorithms make use of known essential and nonessential genes to estimate the probability that the sgRNAs targeting a given gene represent a true dependency.

A well-known cause of variation in CRISPR systems is the variable on-target efficacy of individual sgRNAs. Given multiple screens with the same library, one can attempt a more sophisticated approach where the efficacy of different sgRNAs is inferred directly from the data and thereby estimate gene fitness effects with greater weight placed on the more efficacious sgRNAs. This approach forms the basis of MAGeCK-MLE [12], CERES [13], and JACKS [14]. These approaches are similar in that they treat the log of the relative change in sgRNA abundance during the experiment (log fold change) as the product of sgRNA efficacy and true gene fitness effect, although they employ different statistical assumptions and methods.

To remove bias related to DNA cutting toxicity, CERES uses a nonlinear model to estimate the fitness effects resulting from multiple DNA cuts and infers this relationship for each cell line using its measured copy number profile as input [13]. CRISPRCleanR (CCR) is an unsupervised approach for genome-wide screens: reagents are arranged according to their targeted location on the genome, and regions of systematic guide enrichment or depletion are shifted to the global mean on the assumption that they reflect a copy number alteration [15]. CCR is a "pre-hoc" method that should be used prior to the inference of gene fitness effects. Weck et al. introduced a pair of pre-hoc methods for copy number correction [16], with a local drop out method similar to CCR and a generative additive model method similar to the CERES model of CN effect. Wu et al. introduced an update to MAGeCK-VISPR that similarly uses a paired approach: linear regression with a saturation value for cell lines that have CN profiles and a CCR-style alignment for cell lines that do not have CN profiles [17].

For analyses that compare gene essentiality estimates across screens, variation in screen quality can lead to significant biases [4, 18, 19]. To address this, Boyle et al. introduced a method for identifying and removing principal components that reflect screen quality biases based on observed gene fitness effects of known non-essential genes [19]. A related approach is used to remove screen-quality-related principal components in the CERES data for the Cancer Dependency Map (DepMap) project [4].

Although these methods address individual confounders in CRISPR screens, no existing method addresses all of them. Additionally, CRISPR screens are confounded by incomplete penetrance of the gene knockout phenotype. Given a single late time point measurement, poor knockout of a highly essential gene and complete knockout of a weakly essential gene may result in equivalent depletions, although the true phenotype is very different [20]. This ambiguity can be resolved by measuring fitness at multiple late time points, and with the falling costs of sequencing, an increasing number of experiments do so [21, 22]. Marcotte et al. developed siMEM for combining multiple time points in the context of RNAi experiments [22]. However, siMEM assumes sgRNA dropout occurs exponentially in time. This is a poor fit for CRISPR screens in which some clones escape gene knockout completely. For sgRNAs targeting essential genes, clones with intact function will eventually account for almost all reads and dropout will saturate at a finite value.

To simultaneously address these known challenges, we developed Chronos, an explicit model of cell population dynamics in CRISPR knockout screens. Chronos addresses sgRNA efficacy, variable screen quality and cell growth rate, and heterogeneous DNA cutting outcomes through a mechanistic model of the experiment. Like MAGeCK, but different from other approaches, Chronos also directly models the readcount level data using a more rigorous negative binomial noise model [23], rather than modeling log-fold change values with a Gaussian distribution as is typically done. Copy number related biases are removed using an improved model that accounts for multiple sources of bias. We find that Chronos outperforms other methods on most benchmarks with a single late time point and improves performance considerably when used to model data with multiple time points.

## Results

### Model

Chronos is a mechanistic model designed to leverage the detailed behavior of pooled CRISPR experiments to improve inference of gene essentiality. It models the observed sgRNA depletions across screens and time points to determine the effect of gene knockout on cell growth rate, along with other parameters. CRISPR knockout (KO) of a gene is an inherently stochastic process. Some sgRNAs will fail to cut their target [24]. In the event that they succeed and double-stranded break repair results in an insertion or deletion, in-frame mutations occur in about 20% of cases and may leave protein function intact [5]. Thus, the result of introducing an individual sgRNA reagent in a population of Cas9 positive cells is heterogeneous, with outcomes including total loss, heterozygous loss, partial loss of function mutations, or completely conserved function [25]. The Chronos model simplifies this range of outcomes to a binary pair of possibilities: total loss of function or no loss of function (Fig. 1a). Cells in the latter group will continue proliferating at the original, unperturbed rate. Those in the former will proliferate at some new rate reflecting the effects of a given gene perturbation on cell growth, which is typically the desired readout from the experiment. Concretely, for an sgRNA $j$ targeting gene $g$ in cell line $c$, we model the number of cells $N_{cj}$ with the sgRNA at time $t$ after infection as

$$N_{cj}(t) = N_{cj}(0)\left(p_{cj}e^{R_{cg}^{*}t} + \left(1{-}p_{cj}\right)e^{R_c\,t}\right)$$

where $t = 0$ is the time of infection, $p_{cj}$ is the probability that the sgRNA $j$ achieves knockout of its target in cell line $c$, $R_c$ is the unperturbed growth rate of the cell line, and $R_{cg}^{*}$ is the new growth rate caused by knockout of the targeted gene in the given cell line. For Chronos, we define gene fitness effect as the fractional change in growth rate $r_{cg} = R_{cg}^{*}/R_c - 1$. Gene fitness effects are the primary desired output for this type of experiment.

A wide range of efficacy for sgRNAs in abrogating protein function has been reported [8]. Additionally, we have observed in Project Achilles that screen quality (determined by separation of positive and negative control gene fitness effects) varies substantially across cell lines, due to variable Cas9 activity or other factors [4]. We therefore approximate the knockout probability per sgRNA and cell line as the product of a per-

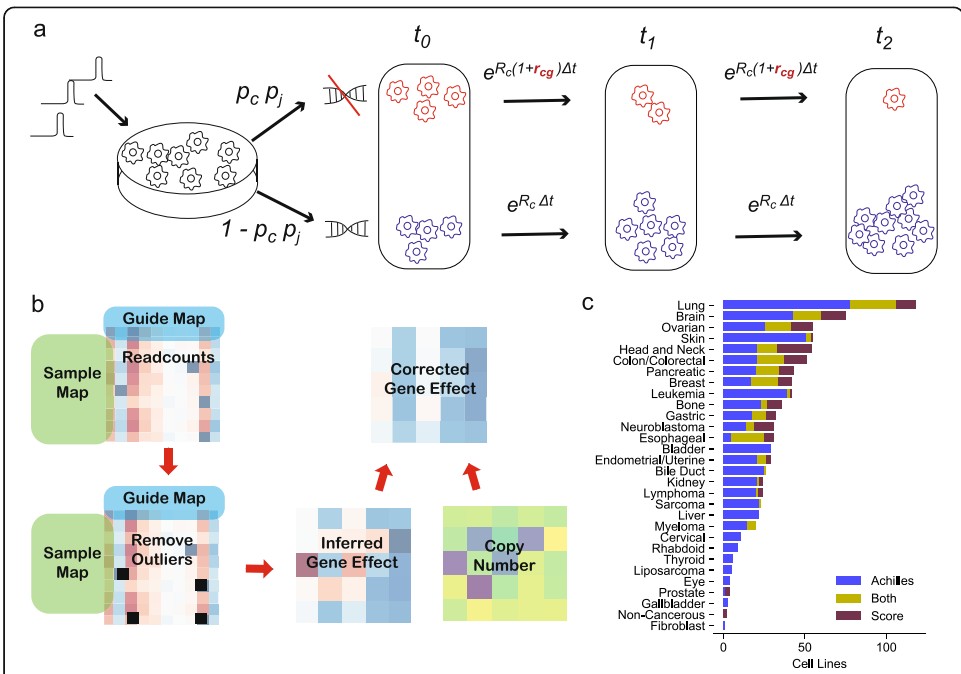

**Fig. 1** Overview of Chronos. **a** Illustration of the model for the simplified case where an sgRNA *j* is introduced targeting essential gene *g* in cell line *c*. Stochastic double-strand repair divides the population of infected cells into two groups, one with (top) and one without (bottom) successful gene knockout, which proliferate at different rates. Successful knockout probability is the product of a per-sgRNA probability $p_j$ and a per-cell line probability $p_c$. The population of cells measured at each subsequent time point is a mix of these two populations. Chronos infers the relative change in growth rate $r_{cg}$. **b** Typical workflow of a Chronos run. Readcounts driven by outgrowing clones are removed, then gene fitness effects inferred using the readcount matrix, sequence map, and guide map. The inferred gene fitness effects are then corrected for copy number effects. **c** Number of cell lines in each lineage for the Achilles and Project Score datasets used for comparison

line and per-sgRNA factor, both constrained to the interval [0, 1]: $p_{ci} = p_c\, p_i$ . The model also includes a gene-specific delay $d_g$ between infection and the emergence of the knockout phenotype, measured in days. Finally, pooled screen sequencing data does not measure the number of infected cells $N_{ci}$ directly but only the proportion of all reads that map to a particular sgRNA. This we assume has an expected value equal to the proportion of cells with that sgRNA : $\langle n_{cj} \rangle = N_{cj}\, /\, \sum_i N_{ci}$. Let the Chronos estimation of $\langle n_{cj} \rangle$ be $v_{cj}$. Then,

$$v_{cj}(t) = Z_{cj}(t) / \sum_j Z_{cj}(t)$$

where

$$Z_{cj}(t) = v_{cj}(0) \left( 1 + p_c p_j \left( e^{R_c r_{cg}\left(t - d_g\right)} - 1 \right) \right) \forall t \geq d_g$$

$$Z_{cj}(t) = v_{cj}(0) \quad \forall t < d_g$$

The parameters are estimated so as to maximize the likelihood of the observed read-counts under a negative binomial distribution, as detailed in the "Methods" section. CRISPR screens occasionally exhibit unexpectedly large readcounts in individual sgRNAs which are suspected to be the result of individual clonal outgrowths unrelated

to the intended CRISPR perturbation [20]. Chronos includes tools to identify and remove suspected outgrowths from read count data and to remove copy number (CN) biases from the inferred gene fitness effect (as described further below). A typical workflow proceeds as shown in Fig. 1b. Note that the efficacy terms $p_c$ and $p_j$ in Chronos directly represent the probability of achieving functional knockout with a given sgRNA in a given cell line, rather than a simple multiplicative scaling of gene scores as in existing models. Additionally, the parameters $p_c$ and $R_c$ together account for much of the variation in screen quality which is a major source of confounding in genomics screens [4, 18, 19]. As a result, Chronos does far better at aligning good and poor-quality screens than competitors (Additional file 1: Fig. S1a): without scaling, the standard deviation of the median of essential gene effects within cell lines in Chronos is 58–79% lower than its competitors. Any algorithm can improve its alignment by scaling the data after fitting, for example forcing the median of common essential gene scores to be –1 in each cell line [4]; however, one cost of doing so is an expansion in the noise of the lowest-quality screens (Additional file 1: Fig. S1b). Further, scaling does not eliminate biases related to variable screen quality, and in some cases can exacerbate these biases (Additional file 1: Fig. S1c).

## Comparison

We first evaluated Chronos on the two largest public datasets of human genome-wide CRISPR screens: projects Achilles [26] and Score [27] containing data from 769 and 317 screens respectively (Fig. 1c). Among the many possible algorithms and combinations of algorithms, we selected three for comparison to represent the space of available choices. Behan et al. [27] used the combination of CCR [15] for copy number correction and BAGEL [10] for gene effect estimation with the first release of Project Score results; we have used the updated pipeline here, substituting BAGEL with the recently published BAGEL2 [11]. We will refer to this pipeline simply as BAGEL2. We chose MAGeCK-MLE (MAGeCK) to illustrate an algorithm which estimates guide efficacy and corrects for copy number while using a more statistically sound negative binomial model for screen noise [12]. Finally, we chose CERES as a method that combines guide efficacy estimates and copy-number correction with information sharing across cell lines via a hierarchical prior on the gene fitness effects [13], which is currently the standard algorithm for the Broad Dependency Map. To focus on the results produced directly from the algorithms themselves, we did not perform any of the batch corrections described in Dempster et al. [4] or Pacini et al. [28] However, for most results, we did normalize the CERES and BAGEL2 data so the median of common essential gene fitness effects in each cell line is –1, as described in Meyers et al. [13]. Some MAGeCK-processed cell lines had control separation too low for this to be a reasonable strategy (Additional file 1: Fig. S1b); we opted to only scale the data globally for MAGeCK.

## Separation of global control genes

The most straightforward indicator of success for a method is the separation it achieves between control sets of known essential and non-essential genes. We used a predefined set of common essential genes, which were identified from independent data, as

positive controls [26]. For each cell line, we used genes that are not expressed in that line as negative controls. The overall distributions of positive and negative control gene fitness effects are shown in Fig. 2a. We measured control separation in four ways, in increasing order of abstraction. First, we computed the null-normalized median difference (NNMD) for all gene fitness effect scores (Fig. 2b) and for each individual cell line (Fig. 2c). It reports the difference between the medians of the control sets normalized by the median absolute deviation of the negative controls. Alone of the measures used here, NNMD can be altered by rank-preserving transformations of gene scores. Chronos outperformed CERES by 38% (ratio of cell line means) in Achilles (paired Student's *t* test $p = 5.9 \times 10^{-275}$, $N = 767$) and 39% in Project Score ($p = 1.6 \times 10^{-110}$, $N = 317$), which in turn outperformed the other two methods.

We next considered an estimate of how many false positive gene fitness effects would be identified by each method. We counted the total number of instances of unexpressed genes scoring in the 15% most depleted gene scores of a cell line (Fig. 2d,e),

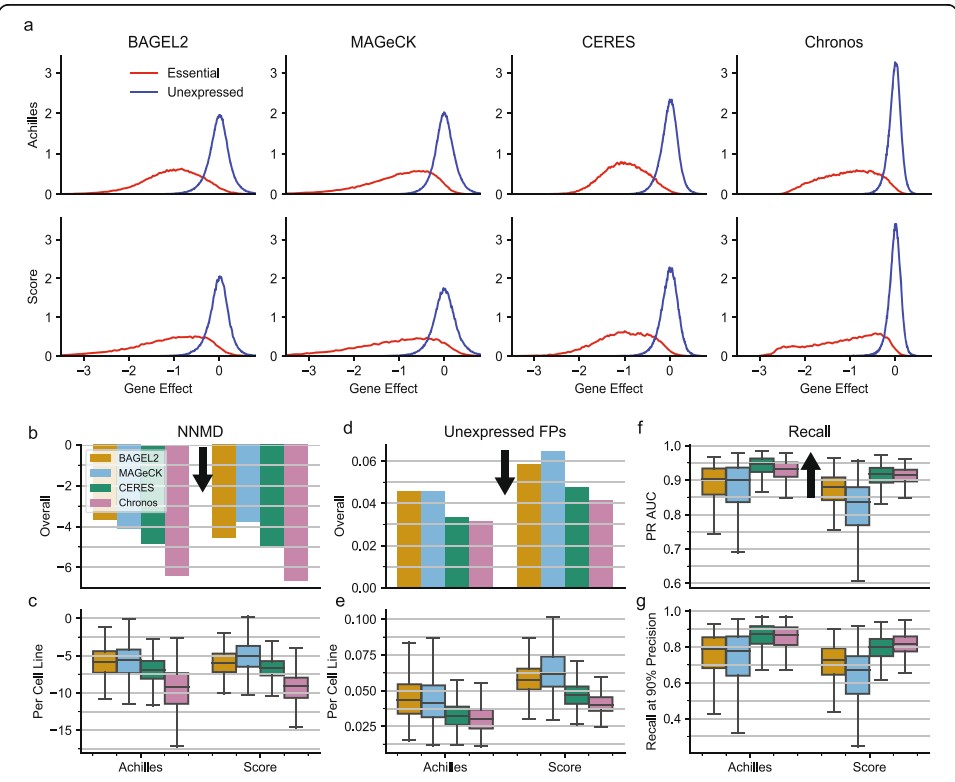

**Fig. 2** Comparison of positive-negative control gene separation across methods for Achilles and Score datasets. **a** The distribution of all gene fitness effects for unexpressed (negative control) genes and common essential (positive control) genes. Unexpressed genes are identified individually for each cell line (Methods). **b** Global separation (pooled across cell lines and genes) between gene scores for common essential genes and unexpressed genes in the cell line where they are unexpressed. Separation computed using null-normalized median difference (NNMD). More negative values indicate stronger separation. Black arrows indicate the direction of improved performance. **c** NNMD for individual cell lines. Boxes indicate the interquartile range (IQR) of NNMDs. Whiskers extend to the last point falling within 1.5 x IQR of the box, and lines indicate medians. **d** Estimated false positive rate, based on the total percentage of unexpressed genes scoring in most depleted 15% of gene scores within cell lines. **e** Fraction of unexpressed genes scoring as false positives in individual cell lines. **f** Area under the precision/recall curve (PR AUC), where recall is the number of common essential gene scores that can be recovered at a given precision. **g** Fraction of possible common essential gene hits identified at 90% precision in individual cell lines

where 15% was chosen because previous estimates indicate that about 15% of genes are true dependencies in a given cell line [4, 29]. Chronos outperformed CERES by 7.2% in Project Achilles ($p = 4.0 \times 10^{-40}$) and 15% in Project Score ($p = 1.1 \times 10^{-31}$).

We then considered the number of known common essential genes that can be recalled as hits with a given precision in each cell line. We measured the total area under the recall/precision curve (PR AUC) for identifying all common essential gene scores (Fig. 2f) and the number of common essentials that can be recovered in each cell line at 90% precision (Fig. 2 g). In all cases, Chronos and CERES outperformed BAGEL2 and MAGeCK. CERES demonstrated a slight but significant advantage of 1.3% by PR AUC in Achilles (paired $t$ test $p = 1.1 \times 10^{-74}$, $N = 762$). However, Chronos recalled on average 2.5% more essentials than CERES in Project Score ($p = 4.2 \times 10^{-8}$, $N = 241$). Other differences between Chronos and CERES were not significant for these metrics.

### Improvement with multiple time points

The preceding analyses are performed entirely on datasets for which there is only one late time point measured per library. We next considered what advantages Chronos could offer when able to exploit sgRNA abundance measured across multiple time points. For this analysis, we used the study by Tzelepis et al. [21] which contains CRISPR KO results for the cell line HT-29 measured at days 7, 10, 13, 16, 19, 22, and 25 post-infection. CERES is not designed to run on a single cell line, and thus was not included in the comparison. Again, we used the metrics NNMD, unexpressed false positives, and PR AUC to assess positive versus negative control separation by Chronos when supplied differing numbers of time points in all possible combinations. Figure 3 illustrates the effect of adding additional time points on each of these three metrics of control separation. All three metrics showed improvement with increasing numbers of time points for all methods, but Chronos showed the greatest improvement. To verify that the Chronos population dynamics model effectively utilizes the data from multiple time points, beyond the simple denoising effect that might be expected from combining biological replicates, we compared the benefit of providing multiple time points simultaneously versus running Chronos separately for each of the same time points individually and taking the median of the results. For all metrics, Chronos performed better when provided multiple time points, and with three or more time points outperformed all competitors on all metrics. (Chronos's copy number correction requires multiple cell lines to identify common essential genes as described further below, so it was not used here.) We conclude that the Chronos dynamical model is able to exploit time-series data for improved performance beyond what could be achieved by naive averaging.

Since many experiments are run in a single cell line with only one late time point, we also quantified Chronos' performance in this context by running it on each Achilles screen individually. Chronos clearly outperformed its competitors as measured by NNMD, exceeding the second-best algorithm MAGeCK by 66% (related $t$ test $p = 1.5 \times 10^{-255}$; Additional file 1: Fig. S2a), but all methods had similar performance by the number of unexpressed false positives (Additional file 1: Fig. S2b) and precision-recall (Additional file 1: Fig. S2c).

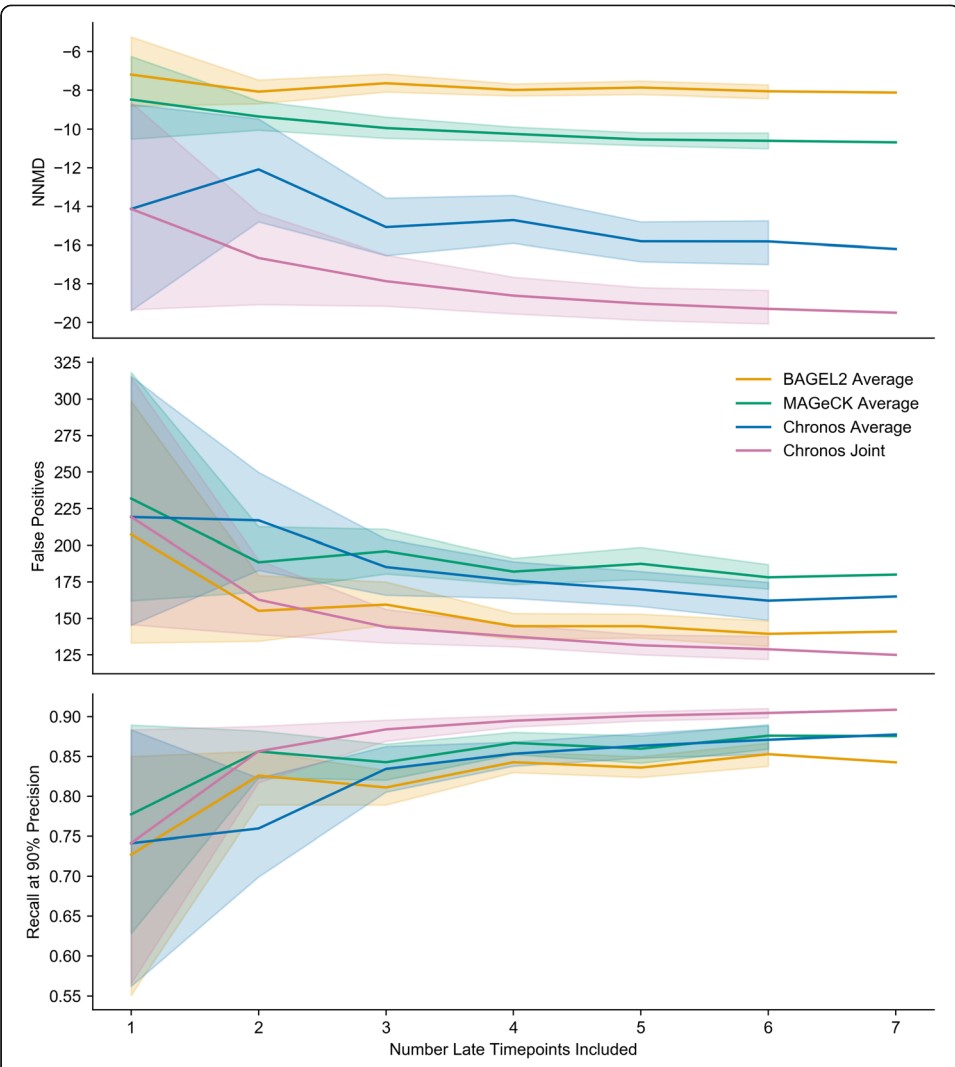

**Fig. 3** Performance improvement with additional time points. The shaded area shows the 95% confidence interval for the (7 choose $n$) possible permutations of $n$ measured time points that can be supplied to an algorithm. "Average" results are the performance achieved by taking a subset of the seven individual runs with a single late time point with a given algorithm and taking the median of their gene fitness effects. Joint results are obtained by running Chronos with a subset of multiple late time points simultaneously

### Identification of selective dependencies

The above analyses largely focus on the ability of each model to differentiate essential and non-essential genes for a given cell line, using a shared set of common-essential genes as positive controls. However, large-scale datasets, such as Project Achilles, are particularly powerful for identifying selective dependencies—genes that have a fitness effect in only a subset of the cell lines—which may represent cancer-selective vulnerabilities. Identifying selective dependencies across screens and cell lines presents a distinct challenge versus recovering known common essential genes. We thus evaluated the performance of algorithms in this context.

To benchmark estimation of selective dependencies, we first analyzed OncoKB genes that exhibit gain- or change-of-function alterations annotated as oncogenic or likely oncogenic [30]. After filtering as described in the "Methods" section, we retained 49

oncogenes known to induce oncogene addiction with activating alterations that had at least one annotated alteration in at least one cell line in at least one dataset. These gene scores were considered selective positive controls in cell lines which had any annotated (likely) gain- or switch-of-function alterations, and selective negative controls in all other lines. There were a median of four cell lines with annotated alterations for each oncogene in the Achilles dataset and two cell lines in Project Score.

To determine how well gene fitness effect profiles for oncogenes were stratified by the known alterations, we measured the difference between cell lines with and without an indicated gain- or switch-of-function alteration using NNMD in each dataset (Fig. 4a). No significant differences were observed between Chronos and the lowest median method due to the small number of oncogenes and small number of positive examples in each gene (paired $t$ test $p = 0.942$, between CERES and Chronos in Achilles, $N = 48$, and $p = 0.511$ in Project Score, $N = 35$). To address this problem, we identified the 43 selective controls that showed signal in any version of the data and pooled the subset that had indicated alterations in the specific dataset to increase statistical power (Methods; Fig. 4b). Chronos outperformed its closest competitor (MAGeCK) in Achilles by 11.2% (permutation testing $p = 0.002$, $N = 1000$). Measuring separation by PR AUC, Chronos outperformed its closest competitor CERES by 10.4% in Achilles ($p = 0.001$, $N = 1000$). BAGEL2 outperformed Chronos in Project Score by 13% according to NNMD, while Chronos was first by 6.5% over MaGECK as measured by PR AUC; however, the differences between first and second-best algorithms in Project Score were not statistically significant.

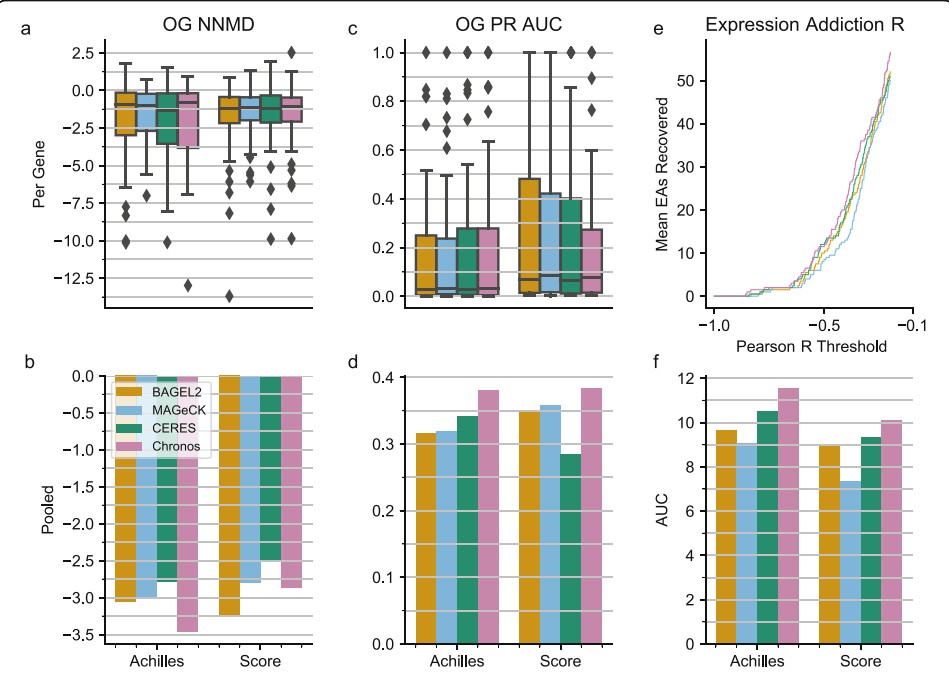

**Fig. 4** Performance on selective dependencies. **a** For each identified oncogene, the NNMD between cell lines with and without the canonical biomarker. **b** The results of **a** when aggregating results across oncogenes with signal. **c** PR AUC for separating cell lines with an indicated alteration from those without for individual oncogenes. **d** The results of **c** when aggregating across oncogenes. **e** The number of known expression addictions (y-axis) found to have a Pearson correlation lower than X (x-axis). **f** The area under the curves of **e** for the individual datasets

To produce a larger set of selective dependencies to evaluate, we used the expression addictions previously identified in Achilles RNAi data [31]. These genes exhibited selective essentiality in cell lines in which they were highly expressed. After subsetting to those present in all algorithms and datasets, and removing any common essential genes (identified from the DepMap releases [26, 32]), we had 106 putative expression addiction genes. We compared the number of potential expression addictions whose gene fitness effects were negatively correlated with their own expression below a given correlation threshold between algorithms, varying the threshold from −1 to −0.2 (Fig. 4e). Chronos consistently identified more expression addiction dependencies as correlated with their own expression at a given threshold than its nearest competitor CERES, resulting in a significantly greater AUC for Achilles (9.0% improvement, two-tailed permutation $p = 2.5 \times 10^{-5}$, $N = 40,000$). Compared to CERES in Project Score, Chronos showed a similar improvement (7.53%) but the difference was not statistically significant due to greater variability in correlation recall ($p = 0.055$; Fig. 4f). Individual genes generally had similar correlations with their expression in all algorithms, with visible systematic improvements but not striking outliers (Additional file 1: Fig. S3-S4).

### Accounting for copy-number biases

It is important for any model for inferring gene KO effects from CRISPR screens to account for the gene-independent DNA-cutting toxicity effect [3]. A precise description of the causes of this cutting toxicity is still lacking, and current literature suggests it may be complex. For example, Gonçalves et al. argue that the copy number effect arises in tandem repeats alone [33]. Furthermore, the relationship between gene fitness effect and copy number varies for different types of genes. For most genes, the observed sgRNA abundance is negatively correlated with copy number across cell lines, but the opposite is true for essential genes (Fig. 5a), likely owing to the decreased probability of achieving complete loss-of-function when more copies of a gene are present. In fact, for essential genes, the average relationship of gene scores with copy number is non-monotonic (Fig. 5b), further highlighting the complexity of the effect. Instead of developing a mechanistic model for the copy number effect, we created a new post-hoc method for correcting copy-number-related biases that can be applied to Chronos gene fitness effects or to analogous output from other modeling approaches. As the copy number effect is a function of both the mean fitness effect of a gene and its copy number in a given cell line, our correction uses a 2D spline model to capture and remove the systematic nonlinear dependence of gene scores on both parameters. Chronos first fits the spline model so that as much of the variance in the gene fitness effects as possible is explained by copy number effect, modulated by the mean gene effect (see Methods). The residuals of the spline model are taken as the corrected gene fitness effects. Where CERES successfully corrects the cutting toxicity effect, it worsens the bias due to incomplete gene KO in common essential genes (Fig. 5c). In contrast, this new correction method incorporated with Chronos corrects both effects simultaneously. The Chronos copy number correction can be run independently on any gene fitness effect matrix with at least three cell lines (to permit inference of mean gene effects).

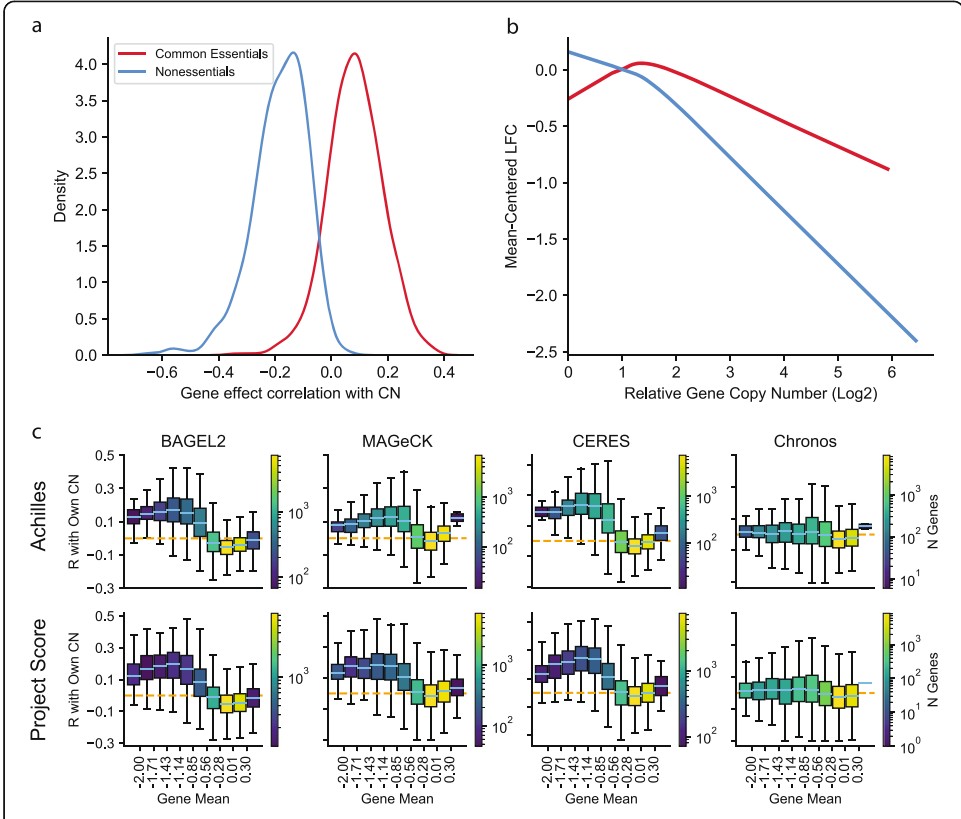

**Fig. 5** The copy number effects. **a** Distribution of correlations for uncorrected gene log fold changes with their own copy number across cell lines in Achilles for common essential and nonessential genes. **b** Lowess smoothed trends for the mean-centered log fold change of known essential and known nonessential genes as a function of copy number. **c** Per-gene correlations of gene fitness effects with its own copy number, binned by mean gene fitness effect. BAGEL2's copy correction is supplied by CRISPRCleanR. The boxes show the IQR for the correlations of genes in the given bin, whiskers extending to the last data point within 1.5 the IQR from the median

## Differences between Chronos and CERES

As CERES is most similar to Chronos in structure and performance, it is worth considering in more detail the differences between the algorithm's estimates. Chronos and CERES generated quite similar profiles across cell lines for most genes (median gene-gene correlations of 0.87 in Project Achilles and 0.85 in Project Score). Many of the genes with lower correlation were strong common essentials, as would be expected from the discussion above of screen quality bias and the differing copy number correction methods, or have little signal to correlate (Additional file 1: Fig. S5). Among the remaining genes, significant disagreement was driven by fundamental disagreement in the reagent-level data, which makes it possible for CERES and Chronos to choose different sets of reagents as reflecting the true knockout signal. Thus, in Achilles, the 3% of genes with less than 0.5 correlation between Chronos and CERES had mean correlation among the LFCs of their sgRNAs of 0.12. The mean sgRNA correlation for the rest of genes was 0.29. In Project Score, the mean sgRNA correlations were 0.19 and 0.33, respectively (Fig. 6a, b). In the general case of sgRNA disagreement, it is difficult to say which of these reagents is exhibiting true on-target signal. However, if only one of four or five sgRNAs shows strong positive or negative signal, it is less likely to reflect true biology. CERES is prone to assigning single outlying sgRNAs high efficacy and

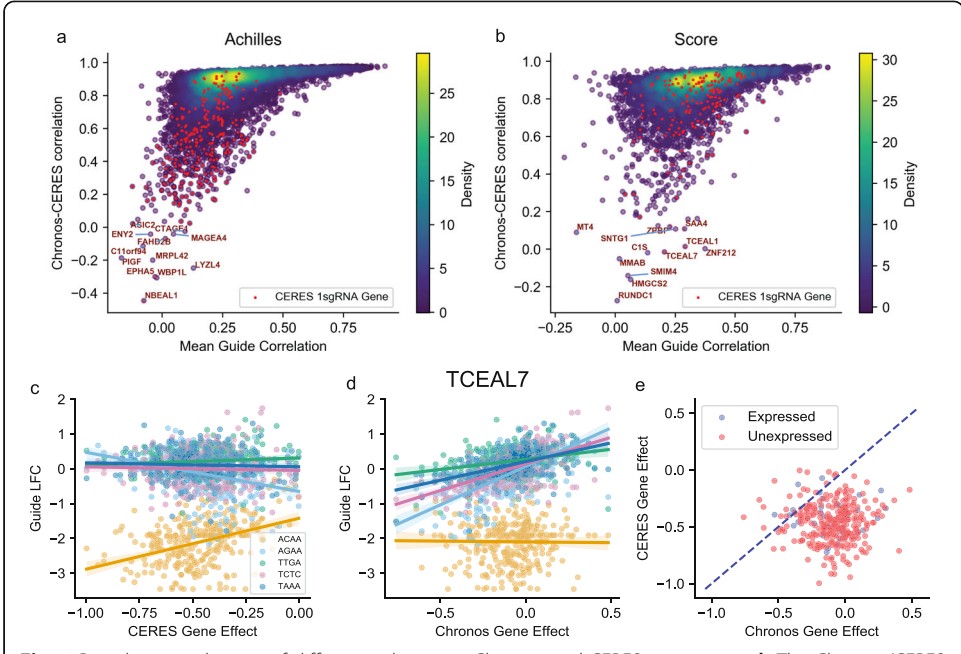

**Fig. 6** Prevalence and cause of differences between Chronos and CERES estimates. **a, b** The Chronos/CERES gene effect profile correlation for genes with Chronos mean effect greater than -0.5 and standard deviation greater than 0.1 (*y*-axis) plotted against the mean correlation of the gene's sgRNAs (*x*-axis). Genes for which CERES estimates a single sgRNA has at least 0.2 greater efficacy than the others are highlighted. **c, d** For the gene *TCEAL7*, the relationship between individual sgRNA log fold-changes (LFCs) and gene effect estimates by either Chronos or CERES. Points are cell lines. Lines show best-fit regression of each sgRNA to the algorithm's gene effect estimate with shaded 90% confidence intervals. Sequence labels of sgRNAs are truncated to the first four nucleotides for clarity. **e** CERES vs Chronos gene effect scores for *TCEAL7*. Dot color indicates whether the gene was expressed or not in each cell line

discounting the contradictory signal from the other sgRNAs in order to produce the best possible fit to observed log fold-changes. In Project Achilles, we identified 475 "1 sgRNA" genes for which CERES assigned a single sgRNA at least 0.2 greater efficacy than all others for the same gene. We found 358 such genes in Project Score. Chronos does not have any such genes.

Relying on the sgRNA with the strongest signal carries risks: sgRNAs with only one exact match may still produce off-target activity at unknown distant sites. In some cases there was clear evidence that the sgRNA CERES selected was acting off-target. An example is transcription elongation factor A likely 7 (*TCEAL7*), a gene unexpressed in most lines. A single sgRNA in the KY library showed appreciable depletion in log fold-change data (Fig. 6c, d). CERES assigned this sgRNA efficacy 1 and reduced efficacy (0.602 and lower) to the other sgRNAs. Consequently, CERES gene effects for this gene in Project Score (but not Achilles) showed significant depletion, including in non-expressing cell lines (Fig. 6e). As Chronos is constrained to always rely on at least two sgRNAs, it did not share CERES' behavior and correctly identified *TCEAL7* as having little effect. Chronos's inhibition against chasing outlying sgRNAs explains some of the reduced false positives found in Chronos compared to CERES (Fig. 2d).

## Discussion
CRISPR functional genomics screens have become indispensable tools in cellular biology. Chronos is a new approach for estimating the fitness effects of gene knockout in

CRISPR screens using a model of the cell population dynamics. We compared the performance of Chronos to existing methods in terms of separation of both global and cell-line-specific essential and nonessential genes. Across two independent large-scale CRISPR screening datasets, Chronos generally outperformed all existing methods we tested, with a few exceptions where it was the second-best method.

A number of factors contributed to improved performance for Chronos even without timecourse data. These are (1) an improved strategy for correcting copy number related bias that produces less distortion in common essential genes, (2) a more sophisticated regularization strategy that reduces bias in estimating selective dependencies while retaining effective information sharing and normalization across screens, (3) an improved generative model of the data at the readcount level, and (4) an explicit model of multiple "screen quality" factors that correct for systematic differences across screens. Additionally, unlike some competitors, Chronos does not assume either a genome-wide experiment [15] or require multiple cell lines [13]. Thus, it is a flexible method that provides improved estimates of gene essentiality from a variety of CRISPR experiments.

An important consideration for CRISPR inference algorithms is the issue of partial mismatches originally raised by Fortin et al. [34]. In some cases, Cas9 appears highly tolerant of one or more mismatches in sgRNA sequence leading to off target effects. The presence of these off-target effects raises the importance of ensuring CRISPR algorithms are robust to single sgRNA outliers. Of the algorithms surveyed, CERES is the most vulnerable to off-target sgRNAs as its structure encourages it to explain the other sgRNAs as inefficacious. In Chronos, a combination of readcount modeling and regularization of sgRNA efficacy effectively prevent it from pursuing single sgRNA outliers, increasing its robustness to off-target sgRNAs. In addition to specific off-target effects, the number of partial mismatches in the genome can also drive a more subtle, broadly distributed depletion in sgRNAs [11, 34]. In our tests, we found that manually correcting this trend before running Chronos had a negligible effect on data quality. We therefore opted to leave out this correction. We note that in the case of TCEAL7, the offending sgRNA has no other exact or single-base-pair tolerant matches in the genome, but does have a number of two- [7] and particularly three- (113) base-pair tolerant mismatches, indicating that any method aiming to predict off-target activity will need to allow for cases where Cas9 tolerates a greater degree of mismatch than considered in Fortin et al.

Chronos has some limitations. Notably, it requires a measurement of pDNA sgRNA abundance, or equivalently a very early time point. In genetic modifier screens one commonly compares treatment vs control at late time points alone [35], and often pDNA data is not collected at all for these experiments. This is easily rectified by including pDNA sequencing in experimental design. Additionally, we evaluated Chronos only for conditions where the majority of cells are proliferating. In the event that most cells are dying—due to a cytotoxic treatment or poor fitness in culture—the model's performance might be adversely affected.

Marcotte et al. suggested that multiple late time point measurements could provide additional value from functional genomics experiments [22]. We found that this is indeed the case for CRISPR experiments. For example, with three late time points, mean performance for Chronos improved by 14% to 31% over its mean performance with only one time point. Importantly, the value of the additional time points is much

greater when integrated with the Chronos dynamical model instead of averaging the individual readouts, indicating that additional time points are not functioning simply as technical replicates but instead are providing useful dynamic information.

## Conclusion

Chronos is an algorithm that uses an explicit model of cell population behavior in CRISPR screens to improve inference of gene fitness effects over the current state of the art. We have implemented Chronos as an open-source python package available at https://github.com/broadinstitute/chronos. Chronos significantly outperformed competitors in multiple tasks across a variety of CRISPR data sets. It performed especially well when provided multiple late time points, indicating that sequencing additional passages can provide substantial benefits in CRISPR experiments.

## Methods

### The Chronos algorithm

Chronos maximizes the likelihood of the observed matrix of normalized readcounts (relative sgRNA abundance) with the following model:

$$v_{cj}{}^L\left(t > d_g{}^L\right) = v_{cj}{}^L(0)\left(1 + p_c{}^L p_j\left(e^{R_c{}^L r_{cg}\left(t - d_g\right)} - 1\right)\right)/Z_c{}^L(t)$$

where

- $c$ indexes cell line, $j$ indexes sgRNA, $g$ indexes gene, $L$ indexes library or batch, and $t$ is the time elapsed since library transduction
- $v_{cj}{}^L(t)$ is the model estimate the normalized readcounts of sgRNA $j$ in cell line $c$ screened in batch or library $L$ at time $t$
- $v_{cj}{}^L(0)$ is the model estimate of the normalized number of cells initially receiving sgRNA $j$
- $p_c{}^L$ and $p_j$ are the estimated CRISPR knockout efficacies in cell line $c$ with sgRNA $j$
- $R_c{}^L$ is the estimated unperturbed growth rate of the cell line
- $r_{cg}$ is the estimated relative change in growth rate for that cell line if gene $g$ targeted by sgRNA $j$ is completely knocked out
- $d_g$ is the delay between infection and the onset of the growth phenotype
- $Z_c{}^L(t)$ is a normalization equal to the sum of the numerator over all sgRNAs $j$ in the cell line for the given library and time point

Unless simultaneously fitting Chronos to data generated from multiple libraries, $L$ can be ignored.

Previous models often treat sgRNAs targeting more than one gene as producing a linear addition of the gene fitness effects in log space. However, it is clear that such a treatment is not biologically supportable [34], and the actual effect of simultaneous knockout is highly dependent on the pair of genes targeted [36]. Consequently, we do not attempt to model multitargeting sgRNAs in Chronos, and our implementation will raise an error stating so if they are encountered.

A naive approach to the distribution of readcounts is a multinomial likelihood. However, biological readcount data often have more dispersion than can be accounted for

by multinomial or Poisson models [12, 23]. The NB2 model is a frequently used parameterization of the negative binomial model that accepts an overdispersion parameter $\alpha$. The likelihood of observing counts $N$ when $\mu$ such counts are expected under the NB2 model is

$$\Pr\ (N|\mu,\alpha) = \frac{\Gamma(N+\alpha^{-1})}{\Gamma(N+1)\Gamma(\alpha^{-1})} \left(\frac{\alpha^{-1}}{\alpha^{-1}+\mu}\right) 1/\alpha \left(\frac{\mu}{\alpha^{-1}+\mu}\right) N$$

We will treat $\alpha$ as a fixed hyperparameter and the observed readcounts as given, so only terms involving $\mu$ need to be retained. Thus, the negative log likelihood becomes

$$\lambda = (N+\alpha^{-1})\ln(1+\alpha\mu) - N\ln\mu$$

Note that an equal change in the scales of $N$ and $\mu$ is equivalent to a change in the scales of $\alpha$ and $\lambda$ up to a constant. With an added constant to ensure the cost is zero when $N = \mu$ and some slight abuse of notation, the core cost function for Chronos reads

$$C = \sum_L \sum_{j\in L} \sum_{c\in L} \sum_{k\in c} \sum_{t\in L} \left(10^6\, n_{kjt}^L + \left(\alpha_c^L\right)^{-1}\right) \ln\left(\frac{1+10^6\,\alpha_c^L\,v_{cj}^L(t)}{1+10^6\,\alpha_c^L\,n_{kjt}^L}\right) - N\ln\left(\frac{v_{cj}^L(t)}{n_{kjt}^L}\right)$$

where $k$ is a replicate of a cell line screened and $t$ both indicates the actual time elapsed since infection and indexes the readout at that time. Rather than estimating library size, we have simply fixed this cost to take in readcounts normalized to reads per million. We made this decision because there is not much evidence that the number of total readcounts found in a screen governs the statistics for the screen. Instead, users can supply the overdispersion parameter $\alpha$ either as a global hyperparameter or as an estimate per cell line and library using independent tools such as edgeR [37]. In our experience, using edgeR estimates of the overdispersion resulted in values so high for some cell lines that they effectively contributed nothing to the cost, despite having clear indications of signal. We opted instead to set a global value.

**Constraints and regularization**

As written, the Chronos model is not identifiable. For example, one can multiply a value $R_c$ by some value and divide all the corresponding values of $r_{cj}$ by the same amount and produce exactly the same estimate. More subtle degeneracies exist between knockout efficacy and gene fitness effect and between gene fitness effect and the estimate of initial cell abundance $v_{cj}{}^L(0)$. Even for terms not exactly degenerate, the quality of the model fit is improved by regularization. We applied the following constraints and penalties:

$v_{cj}{}^L(0)$: This initial time point could in theory be inferred in the same way as any other parameter, with pDNA abundance supplied as a $t = 0$ time point. However, previous work has found evidence of some systematic errors in initial sgRNA abundance as measured in pDNA (13) in which some sgRNAs appear to be consistently more or less abundant in cell lines than could be explained by either target knockout effect or the measured pDNA abundance. Additionally, some experiments batch pDNA measurements across many screens. Thus, we instead use the following:

$$v_{cj}{}^L(0) = n_{b(c)j\,0}{}^L \cdot exp\left(\rho_{bj}{}^L\right)$$

where $n_{b(c)j\,0}{}^L$ is the median measured relative abundance of sgRNA $j$ in the pDNA batch $b$ of cell line $c$ in library $L$ and $\rho_{bj}{}^L$ is a parameter to be estimated. It is constrained to have mean 0 for the complete set of sgRNAs that target any specific gene (recall that Chronos does not accept sgRNAs annotated as targeting multiple genes). Additionally, it is subject to the penalty

$$C_\rho = \chi_\rho \frac{1}{K} \sum_L \sum_{b \in L} \sum_{ij \in L} \left(\rho_{bj^L}\right)^2$$

where we will use $K$ here and elsewhere to indicate the total number of terms in the sums and $\chi_\rho$ is a regularization hyperparameter equal to 1.0 by default.

$p_c{}^L$: In the case of only one late time point, this cell line screen quality parameter is degenerate with both the unperturbed cell line growth rate $R_c{}^L$ and (in global scaling) with the sgRNA quality parameter $p_j{}^L$. To avoid these degeneracies, we estimate this parameter directly from the fold change in the $n$th most depleted sgRNA at the last available time point for the library, where $n$ is the 99th percentile of sgRNAs by default.

$p_j$: As we noted above, sgRNA efficacies in CERES tend to amplify the most extreme sgRNA results. Specifically, if three of four sgRNAs show little depletion in every line and one sgRNA shows substantial depletion in any cell lines, CERES will usually assign high efficacy to the outlying sgRNA and low efficacy to the others. To prevent Chronos from similarly chasing a single sgRNA, we force it to assign efficacy values near 1 to at least two guides with the following term:

$$C_\rho = \chi_\rho \frac{1}{K} \sum_g \sum_{i \in g} I_{gj} P_j{}^{-1}$$

where $\chi_\rho$ is a regularization hyperparameter set to 0.5 by default and $I_{gj}$ is an indicator function with value 1 iff the sgRNA $i$ is currently estimated to be the first or second most efficacious sgRNA for the gene $g$.

$R_c{}^L$: The per-cell line and library unperturbed growth rate is degenerate with the cell efficacy and the individual rows of $r_{cg}$. We constrained it to have positive values and mean 1. Additionally, it is regularized with the penalty

$$C_R = \chi_R \frac{1}{K} \sum_L \sum_{c \in L} \ln R_c^L$$

where $\chi_R$ is a regularization hyperparameter with default value 0.01. The log penalty is chosen because it has little influence on a parameter constrained to have mean 1 unless some cell lines approach 0, a behavior we have observed occasionally with small internal datasets.

$r_{cg}$: The gene fitness effect matrix is regularized in two ways. First, the global mean is strongly regularized towards 0:

$$C_{r1} = \chi_{r1} \sum_c \sum_g r_{cg}$$

where $\chi_{r1}$ is a regularization hyperparameter with default value 0.1. The second regularization is a smoothed hierarchical kernel prior

$$C_{r2} = \frac{1}{K} \sum_c \sum_g \sum_h \left( \kappa_{g-h} (r_{ch} - \bar{r}_h)^2 \right)$$

where $\kappa_{g-h}$ is a kernel function of the rank distance of the means of the effects of genes $g$ and $h$, and $\bar{r}_h$ is the mean value of gene $h$ across cell lines. The kernel is a combination of two terms:

$$\kappa_{g-h} = \chi_h \delta_{gh} + \chi_k \frac{1}{b} \ exp\left((g-h)^2 / 2\sigma^2\right)$$

where $\chi_h$, $\chi_k$ and $\sigma$ are hyperparameters with default values 0.1, 0.25, and 5, $\delta_{gh}$ is the Kronecker delta, and $b$ is a normalization term that makes the sum of the Gaussian exponential 1. For computational efficiency, the support of $\kappa$ is restricted to $3\sigma$ in either direction from zero. For interpretability, we recommend users shift and scale the whole inferred gene matrix so the median of all nonessential gene scores is 0 and the median of all essential gene scores is -1 *globally*, not per cell line.

$d_g^L$: In theory, the gene knockout phenotype delay parameter could be inferred given sufficient time points, in particular time points measured relatively close to infection. In practice, we have never found a benefit to trying to do so. Instead, it is fixed as a hyperparameter with default value 3 days, which the authors have found roughly approximates the onset of a fitness phenotype for many essential gene knockouts.

### Implementation and fitting

Training Chronos consists of choosing the free parameters to minimize the combined cost function

$$C_T = C + C_\rho + C_p + C_R + C_{r1} + C_{r2}$$

We implemented the Chronos model in tensorflow v1.15 and used the native AdamOptimizer to train the parameters. The pDNA offsets $\rho_{bj}^L$ are initialized to 0, the guide efficacies $p_j$ are initialized to values near 1 with a small random negative offset, unperturbed growth rates $R_c$ are initialized near 1 with a random Gaussian offset (standard deviation 0.01), and for the gene fitness effect $r_{cg}$ the gene means and the per-cell-line scores are separately initialized to very small random values in the interval $[-10^{-4}, 0.5 \times 10^{-4}]$.

We use a few tricks to improve learning performance. First, to ensure numerical stability in the exponents, time values are multiplied by a constant with default value 0.1, equivalent to measuring time in units of 10 days. Second, the core cost $C$ is rescaled so it has an initial value of 0.67 by default, equivalent to renormalizing the hyperparameters. This ensures more consistent effects for the regularization terms in datasets with different sizes. With this adjustment, we have found that the default hyperparameters work well for all cases tested, including sub-genome libraries with small numbers of screens. Third, we find that using a fixed learning rate for AdamOptimizer causes either an initial explosion in the cost function before the optimizer begins minimizing, or else inefficient learning if the rate is very small. The final optimal parameters learned for both these cases are quite similar, but learning is inefficient. We therefore initialize the optimizer with a default maximum learning rate of $10^{-4}$, rising exponentially over a burn in period of 50 epochs until it reaches a default value of 0.02.

Finally, we optimize the gene fitness effect alone during the first 100 epochs. Because the core cost function is convex when gene fitness effect alone is considered, this helps ensure stability for the optimum found by Chronos. Training runs for a number of epochs specified by the user (default 801).

We have noticed a pattern of rare clonal outgrowth in CRISPR data, where a single sgRNA for a single replicate of a cell line will show unusually large readcounts. This may be due to a random fitness mutation or other artifact. We provide users an option to preprocess their readcount data by identifying and removing these rare events. Specifically, for cases where the log2 fold change of a guide in a biological replicate is greater than a specified threshold, and the difference between that log fold change and the next highest log fold change for an sgRNA targeting that gene in the given replicate is lower by a specified amount, the outgrower is NAed. About 0.02% of all readcount entries in Achilles and Project Score were NAed for suspected clonal outgrowth.

### Removing copy number effect

To remove copy number bias, Chronos provides a function that accepts a matrix of gene-level copy number (processed as in DepMap: $\log2(x + 1)$ where $x$ is the relative copy number) and a matrix of gene fitness effect. It constructs a two-dimensional cubic spline representation for each gene fitness effect score in each cell line, where the first dimension is the copy number of the gene in a given line and the second is the mean effect of the gene across lines. By default, the model uses ten knots linearly spaced in copy number space and five knots in mean gene fitness effect space, spaced pseudo-exponentially (meaning, exponential if the first percentile mean gene fitness effect is taken as 1 and the other mean gene fitness effects shifted accordingly), so that knots are more concentrated in the region of strong negative mean gene fitness effect where the copy number effect changes fastest. The model for the copy number effect is then written as follows:

$$y_{cg} = w_c \sum_k \theta_k B_{cgk}$$

where $B_{cgk}$ is the spline basis representation, $\theta_k$ are the spline coefficients, and $w_c$ is a per-cell-line parameter in the interval (0, 1] that weights the strength of the copy number effect in that line. The model minimizes the cost function

$$C_{CN} = \sum_{c,g} \left( r_{cg} - y_{cg} \right)^2 + X_w \sum_c \ln (w_c)^2$$

then returns the residuals, which can be treated as bias-corrected gene fitness effect estimates for downstream analysis.

### Algorithm runs

For all algorithms, we began with the readcount tables provided for Achilles [26] and Project Score [27], which have each undergone different QC pipelines.

To compute log fold change (LFC), we calculated the base-2 log of reads per million (RPM) + 1 and subtracted the pDNA values for the appropriate batch from the late time points. For Achilles data, which has multiple pDNA measurements, we summed pDNA measurements from the same batch prior to computing RPM. To calculate a

naive gene fitness effect score for Fig. 5a, b, we filtered out all sgRNAs with multiple gene targets and computed gene fitness effects per replicate as the median of the LFCs for all sgRNAs targeting the gene in question. We then computed cell line gene fitness effects by taking the median of replicate gene fitness effects.

All gene effect estimates were normalized (shifted and scaled) globally so that the median of the medians of reference nonessential genes was 0 and the median of the medians of essential genes was −1 across cell lines. This was done for visual interpretability only and has no effect on the metrics evaluated in this manuscript.

The Bagel2-CRISPRcleanR pipeline was run on each dataset using the run_bagel_crisprcleanr.py script from version 2 build 115 of the BAGEL software, which called a local installation of version 2.2.1 of the R package CRISPRcleanR. The provided core essential (CEGv2.txt), non-essential (NEGv1.txt), KY library ("Yusa" flag), Avana library (Avana_library_forCRISPRcleanR_hg38.txt), KY alignment (KYv1_align_summary.txt), and Avana alignment (Avana_align_summary_hg38.txt) files were used, as well as the default bootstrapping and normalization options, with multi-targeting correction enabled. The readcounts and screen info files were provided as inputs to the pipeline. To prepare the readcounts, we summed pDNA batches with multiple measurements. The screen info file mapped cell lines to their replicates and pDNA batches, with cell lines made up of replicates from different pDNA batches split and treated as separate samples. The pipeline outputted Bayes factor files (.bf) for each cell line, which were then concatenated to create a gene essentiality matrix for each dataset. Finally, the Bayes factor profiles for the aforementioned cell lines split into multiple samples were averaged.

For Chronos, sgRNAs were filtered similarly. Suspected clonal outgrowths were removed with the methods described above and the model trained for 801 steps. The resulting gene viability effect matrix was then globally normalized as described above and then copy number corrected as described above using the DepMap gene level copy number data [26]. Genes present in the gene viability matrix and not the copy number matrix were assumed to have normal ploidy (assigned value 1).

CERES results were taken directly from the Dependency Map file gene_effect_unscaled [26, 32].

To run MAGeCK-MLE, we filtered the sgRNAs as described above. These cleaned tables were then split by pDNA batch. Due to issues with memory consumption (greater than 50GB) and run time (greater than 3 days) for larger batches, batches which consisted of more than 70 cell lines were further subdivided into sets of approximately 50 cell lines. Design matrices were constructed to indicate pDNA as the baseline and map experimental replicates to their respective cell line or time point. MAGeCK 0.5.9 was run with the "mle" subcommand in order to perform maximum-likelihood estimation of the gene essentiality (beta) scores. The sgRNA efficiency was iteratively updated during EM iteration using the "--update-efficiency" flag. In order to further reduce memory usage and runtime, permutation was performed on all genes together, rather than by sgRNA count, using the "--no-permutation-by-group" flag and genes with more than 6 sgRNAs, a total of 33 in the Achilles and 673 in Project Score, were excluded from the maximum-likelihood estimation calculation using the "--max-sgrnapergene-permutation" parameter. The effects of copy number variation were corrected using the "--cnv-norm" parameter. The DepMap gene level copy number data [26] was converted to log2 and genes with missing data were imputed with zeros, indicating no change in

copy number. The beta scores were then extracted from the outputted gene summary files and aggregated across batches to construct a gene fitness effect matrix for each dataset. Batch effects between the cell line batches were removed by ComBat [38].

Runs for the single cell line analysis were performed in Chronos as described above for the full Achilles dataset, but filtering the read count matrix to include only one cell line at a time during inference. Gene effects from the runs for individual cell lines were stored and concatenated afterwards into a single matrix. A similar approach was used with MAGeCK. Neither Chronos nor MAGeCK single line runs were copy-number corrected.

## Analysis methods

Unless otherwise reported, all *p* values were calculated using the python scipy.stats package using the given test, and all analyses were restricted to genes in common across all versions of the data.

For each dataset, algorithm, and cell line, the NNMD score was calculated as the difference in the medians of positive controls and negative controls, normalized by the median absolute deviation of negative controls. Positive controls were taken as the intersection of common essential genes in the DepMap public 20Q2 dataset with the genes present in all datasets and algorithms. In each cell line, negative controls were unexpressed genes: those with expression less than 0.01 in that cell line according to the DepMap public 20Q2 dataset. When such genes fell in the bottom 15% of all gene fitness effect scores for the cell line for a given algorithm and dataset, the score was considered an unexpressed false positive.

Precision and recall for global controls were estimated directly from the gene fitness effects of common essential and unexpressed genes in each cell line individually using the sklearn.metrics function precision_recall_curve.

To produce a list of selectively essential genes, we downloaded the OncoKB annotations for all variants (http://oncokb.org/api/v1/utils/allAnnotatedVariants, accessed July 15, 2020) [30]. Initially, 468 genes were present. Rows were filtered for alterations listed as "(Likely) Oncogenic," with mutation effects in "(Likely) Gain of Function" or "(Likely) Switch of Function," leaving 225 genes. We filtered out those not present in all versions of the Achilles and Project Score data, leaving 183 genes. We then removed those identified as common essential from the CERES gene effect scores in the DepMap releases for Project Achilles and Project Score, leaving 166 genes. DepMap fusion and mutation calls were used to identify cell lines with the indicated alteration, where the altered gene was treated as a selective positive control. For fusions, both gene members of the fusion were treated as selective positive controls. If no dataset had any cell lines with an indicated alteration for a gene, that gene was dropped, leaving 73 oncogenes. We then conducted a literature review of the remaining 73 oncogenes and removed (1) genes where the annotated alterations do not sufficiently capture the conditions where the gene becomes a dependency (for example, genes where the primary indicator of oncogenic status is high expression), (2) genes known to act as tumor suppressors in at least some contexts, (3) genes whose function is expected to be significant only in vivo, (4) genes whose function is chiefly to confer drug resistance, and (5) genes with insufficient evidence of oncogenicity. Forty-nine genes remained.

Reported metrics (NNMD and precision/recall curve) were calculated both per oncogene between cell lines with and without indicated alterations and collectively after pooling all selective positive control and selective negative control gene fitness effects. We pooled only the 43 genes which had NNMD below -0.5 in at least one version of the data. To calculate permutation $p$ values for the pooled case, we randomly switched gene fitness effect scores between first and second-best algorithms, calculated the difference between their metrics, and repeated 1000 times. We then calculated the rank of the absolute value of the observed difference among the absolute values of the permutation differences and used this to determine the empirical $p$ value as described in North et al. [39].

## Supplementary Information

---

**Additional file 1:** Supplementary figures.

**Additional file 2:** Review history.

---

### Acknowledgements
Not applicable.

### Peer review information

### Review history
The review history is available as Additional file 2.

### Authors' contributions
J.M.D. conceived the study, wrote the code, conducted the analyses, and wrote the manuscript. I.B. performed the MAgECK-MLE and CCR-BAGEL2 runs. F.V. curated the list of oncogene addictions. A.T. and J.M.M. supervised the study. All authors read, edited, and approved the final manuscript.

### Funding
This work was funded by The Minderoo Foundation, The Robertson Foundation, and the Cancer Dependency Map Consortium. The funders did not influence the conception, design, or analysis of the study or the writing of this manuscript.

### Availability of data and materials
The code for Chronos is available as a Python package at https://github.com/broadinstitute/chronos [40]. The code is provided under the modified BSD license [41]. The version of Chronos used to generate the datasets in this manuscript has been archived in zenodo [42]. The datasets generated and/or analyzed during the current study and the code used to generate the panels and analyses in this manuscript are available on Figshare at https://doi.org/10.6084/m9.figshare.14067047 [43]. The publicly available datasets used in this study are reproduced in the figshare and include DepMap 20Q2 [26], Project SCORE [27], Tzelepis et al. [21], and OncoKB [44].

## Declarations

### Ethics approval and consent to participate
Not applicable.

### Consent for publication
Not applicable.

### Competing interests
F.V. receives research support from Novo Ventures. D.E.R. receives research funding from members of the Functional Genomics Consortium (Abbvie, Bristol-Myers Squibb, Jannsen, Merck, Vir) and is a director of Addgene, Inc. A.T. is a consultant for Tango Therapeutics, Cedilla Therapeutics, and Foghorn Therapeutics. W.C.H. is a consultant for Thermo-Fisher, Solasta, MPM Capital, iTeos, Frontier Medicines, Tyra Biosciences, RAPPTA Therapeutics, KSQ Therapeutics, Jubilant Therapeutics, and Paraxel.

### Author details
[1]Broad Institute of MIT and Harvard, 415 Main Street, Cambridge, MA 02142, USA. [2]Dana-Farber Cancer Institute, 450 Brookline Ave, Boston, MA 02215, USA.

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

## 