## [**Additional file 2:** Review history. · Genome Biology]

Review History

First round of review

Reviewer 1

Were you able to assess all statistics in the manuscript, including the appropriateness of statistical tests used? Yes, and I have assessed the statistics in my report.

Comments to author:

Dempster et al. present Chronos as a new model for interpretation of CRISPR functional genomics screens. The model's assumptions and performance are solid however the improvement over CERES although significant is somewhat modest for single endpoint screens as there is a high level of correlation between the two approaches for the vast majority of genes. It would build interest in Chronos to show the overlap between Chronos and CERES and then highlight one biological example where CERES missed what Chronos finds.

I found the discussion of the complex nature of copy number correction for essential vs non-essential genes to be very interesting and was not something I had previously considered.

Although I appreciate that Chronos is a more significant improvement for multi-timepoint screens it seems fairly unlikely that most efforts or laboratories at this time will be capable of running time course genome-scale screens given this vastly increases the cost and work associated with any project.

However, given it seems Chronos will be a major feature of DepMap moving forward it is appropriate that Chronos, even as a relatively modest improvement over CERES for single end point screens, is featured in a high quality publication such as Genome Biology as this work will likely be widely cited.

The documentation on github is important as Chronos level of adoption will be dictated by ease of use relative to MAGeCK for individual laboratories.

Reviewer 2

Were you able to assess all statistics in the manuscript, including the appropriateness of statistical tests used? Yes, and I have assessed the statistics in my report.

Comments to author:

Dempster et al present a new method termed Chronos to estimate gene fitness effects from pooled CRISPR effects. The motivation behind the approach is to address a number of known issues with CRISPR screens - differing efficacies for sgRNAs targeting the same gene, variation in copy number causing variable CRISPR-cutting induced toxicity, variable screen quality potentially due to Cas9 expression variation, and variability in loss-of-function effects resulting from different DNA repair outcomes. Different subsets of these issues are already taken into account by existing methodologies (such as CERES, Mageck, CRISPRCleanR) but not all of them. The motivation for Chronos is to address all of these issues primarily by explicitly incorporating their effects into the Chronos model. The authors demonstrate that according to a

number of metrics Chronos outperforms popular existing approaches. The authors have made all data and code used for analysis and generating the graphs available at the linked figshare.

Major comments:

- The authors compare their method to CRISPRCleanR, a method for correcting copy number artefacts in CRISPR screens. However this is, as the authors note, a "pre-hoc" method typically used before a gene scoring methodology is used to aggregate multiple sgRNAs into gene level scores. For example, Project Score, the large scale CRISPR screening effort from the Sanger Institute, uses CRISPRCleanR in combination with BAGEL2 (Kim and Hart 2021) to derive gene fitness scores. A fairer comparison would therefore be between Chronos and the CRISPRCleanR/BAGEL2 pipeline used for Project Score.

- Multiple-gene targeting and mismatch tolerance is a major challenge for the interpretation of gene knockout effects (Fortin et al. 2019) but is not discussed in the main body of the paper. It is noted in the methods that unlike CERES, Chronos does not attempt to model multi-targeting sgRNAs and requires them to be filtered out. However, Fortin et al. showed that guides with perfect matches to non-coding regions, as well as guides that align with a single base pair mismatch can also confound gene fitness effects. BAGEL2 uses CRISPRCleanR for copy number correction but additionally takes non-coding perfect matches and 1bp mismatches of the sgRNAs into account (like Chronos, it discards guides that map to multiple coding regions). At minimum the consequences of these mismatch effects should be discussed.

- The legend and figure for Figure 5 do not match so it is impossible to make sense of this figure.
- Partially because of the issues with Figure 5 it is hard to make sense of the oncogene addiction analysis. It is also unclear what the criteria for selection of oncogenes was - in the methods it says "Finally, the 69 surviving oncogenes were manually curated to identify oncogenes known to induce oncogene addiction, leaving 40 genes". How exactly did the authors manually curate the list of 69 genes?

Overall it seems that Chronos does not perform significantly better than the other approaches at identifying oncogene addiction effects. It would be interesting to see correlations between the Chronos and other metrics for individual genes to see if they are good at distinguishing the effects for the same or different genes (similar to what was done in Fig 4)

- The authors create a 'post-hoc' method for correcting for copy number biases (p22, Figure 6). It's not clear to me if the rest of the figures in the paper (e.g. Fig 2) make use of the Chronos scores with this post-hoc copy number correction. It's important to clarify this as potentially the copy number correction improves / worsens the quality of the results.

Minor comments

- In text (main and methods) and the Fig. 1 legend, i and j are both used interchangeably to refer to a specific sgRNA.
- p16 typo - 'versys'
- p42 dated preprint reference, Goncalves et al is now published in Genome Biology - p43 broken reference to Wu et al

- 'Rare clonal outgrowth' is mentioned in early on (p10) but only explained in the methods, this should be explained earlier on.
- The package on GitHub is well described and has a complete example notebook which will facilitate usability. One additional thing that would be useful to note for future users is an approximate sample runtime for Chronos, e.g. for DepMap 20Q4.

Minor comments on figures:

- It's not possible to visually distinguish between the CERES and Chronos box plots in Fig. 2f, perhaps outliers should not be included in this graph?
- Fig. 2g: y-axis label should read "Recall at 90% Precision" for clarity.
- It would be clearer if Fig. 3 used the same color assignment for different methods as used in Fig. 2
- It's not clear why the arrows to indicate the direction of increasing performance are only used in Fig. 4, after those metrics have already been introduced in previous figures.
- The box plots in Fig 6c could be scaled up a bit, they're very small compared to Fig. 6ab..

Dear Dr. Pang,

Please find attached our revised manuscript. To address the points raised by the reviewers, we have added a new figure (now Figure 6) and three new supplementary figures (3, 4, and 5), along with a new Discussion paragraph and many smaller changes. We also conducted a number of tests and analyses to closely investigate some of the issues raised by the reviewers which did not fit in the scope of the manuscript or the Chronos algorithm, but which we think may be of interest to the reviewers. These are described below where appropriate. Our detailed reply to the reviewers follows.

Sincerely,

Dr. Joshua Dempster

Detailed Response

Reviewer #1: Dempster et al. present Chronos as a new model for interpretation of CRISPR functional genomics screens. The model's assumptions and performance are solid however the improvement over CERES although significant is somewhat modest for single endpoint screens as there is a high level of correlation between the two approaches for the vast majority of genes. It would build interest in Chronos to show the overlap between Chronos and CERES and then highlight one biological example where CERES missed what Chronos finds.

We appreciate the reviewer's suggestion. To address it, in the revised manuscript we explore the relationship between CERES and Chronos scores in substantially more detail -- adding a new section "Differences between CERES and Chronos" with a new main figure 6 and supplementary figures 3, 4 and 5.

As we show in the new Figures 6 and S5, CERES and Chronos show strong agreement for the great majority of genes. Indeed, we expect qualitatively similar results for most genes using any reasonable processing method, and thus we target broad quantitative improvements with Chronos. See for example the new supplementary figures 3 and 4 which illustrate both the systematic improvement with Chronos over existing methods but also the lack of dramatic outliers in relative performance under any of the four algorithms.

We observe dramatic differences between CERES and Chronos only when there is strong disagreement between different reagents for the same gene, in which case it is possible for the algorithms to choose different subsets of reagents to estimate gene effects. Given variable on-target efficacy as well as off-target effects, we do not believe a single method can be guaranteed to always choose the right reagents, and we want to avoid cherry-picking a case where Chronos might have only gotten lucky. However, there is a class of genes where we think Chronos is systematically more likely to make the correct choice: when there is a single outlying sgRNA that disagrees with other sgRNAs targeting the gene. CERES is apt to explain these

cases by assuming the outlier is the only efficacious sgRNA and ignoring the rest. We designed Chronos so it must credit at least two sgRNAs with full efficacy. This allows it to correctly treat cases where the outlier sgRNA is clearly mistaken, such as when the outlier appears to be depleted even when cell lines do not express the gene in question. The gene TCEAL7 in Project Score is a striking example of a case where Chronos avoids a mistake CERES makes. In the new figure 6 we illustrate both the broad point about disagreement between Chronos and CERES being the product of guide-level disagreement, and how CERES' chasing of single outliers can create artificial signal.

I found the discussion of the complex nature of copy number correction for essential vs non-essential genes to be very interesting and was not something I had previously considered.

We thank the reviewer for their interest. Indeed, to our knowledge this point has not been previously raised about the copy number effect in CRISPR screens.

Although I appreciate that Chronos is a more significant improvement for multi-timepoint screens it seems fairly unlikely that most efforts or laboratories at this time will be capable of running time course genome-scale screens given this vastly increases the cost and work associated with any project.

We can say that at the Broad Institute we have observed a number of experiments now opting for multiple time point collection, several of which we expect to be published in the near future. Depending on the design, the cost of setting aside and sequencing cells from intermediate passages is now a fairly small fraction of the total cost of a typical 2D experiment, and the speed of a perturbation's effect can be of high clinical salience. However we do agree that the majority of CRISPR experiments will continue to use single late timepoints in individual cell lines for the foreseeable future, and acknowledge that there will be less difference between methods in this setting.

However, given it seems Chronos will be a major feature of DepMap moving forward it is appropriate that Chronos, even as a relatively modest improvement over CERES for single end point screens, is featured in a high quality publication such as Genome Biology as this work will likely be widely cited.

The documentation on github is important as Chronos level of adoption will be dictated by ease of use relative to MAGeCK for individual laboratories.

We agree about the importance of documentation. One of our goals with Chronos is to increase ease of use relative to CERES. We intend to continue testing and improving Chronos.

Reviewer #2: Dempster et al present a new method termed Chronos to estimate gene fitness effects from pooled CRISPR effects. The motivation behind the approach is to address a number of known issues with CRISPR screens - differing efficacies for sgRNAs targeting the same gene, variation in copy number causing variable CRISPR-cutting induced toxicity, variable

screen quality potentially due to Cas9 expression variation, and variability in loss-of-function effects resulting from different DNA repair outcomes. Different subsets of these issues are already taken into account by existing methodologies (such as CERES, Mageck, CRISPRCleanR) but not all of them. The motivation for Chronos is to address all of these issues primarily by explicitly incorporating their effects into the Chronos model. The authors demonstrate that according to a number of metrics Chronos outperforms popular existing approaches. The authors have made all data and code used for analysis and generating the graphs available at the linked figshare.

We appreciate the reviewer's close reading and thoughtful suggestions.

Major comments:

- The authors compare their method to CRISPRCleanR, a method for correcting copy number artefacts in CRISPR screens. However this is, as the authors note, a "pre-hoc" method typically used before a gene scoring methodology is used to aggregate multiple sgRNAs into gene level scores. For example, Project Score, the large scale CRISPR screening effort from the Sanger Institute, uses CRISPRCleanR in combination with BAGEL2 (Kim and Hart 2021) to derive gene fitness scores. A fairer comparison would therefore be between Chronos and the CRISPRCleanR/BAGEL2 pipeline used for Project Score.

We have replaced the CRISPRCleanR (CCR) results with the CCR-BAGEL2 pipeline suggested by the reviewer. Notably, CCR-BAGEL2 outperforms Chronos on separating oncogene dependencies as measured by NNMD in the Project Score data (Figure 4a, b), albeit not significantly ($p=0.17$). Other aspects of the manuscript remain qualitatively unchanged.

- Multiple-gene targeting and mismatch tolerance is a major challenge for the interpretation of gene knockout effects (Fortin et al. 2019) but is not discussed in the main body of the paper. It is noted in the methods that unlike CERES, Chronos does not attempt to model multi-targeting sgRNAs and requires them to be filtered out. However, Fortin et al. showed that guides with perfect matches to non-coding regions, as well as guides that align with a single base pair mismatch can also confound gene fitness effects. BAGEL2 uses CRISPRCleanR for copy number correction but additionally takes non-coding perfect matches and 1bp mismatches of the sgRNAs into account (like Chronos, it discards guides that map to multiple coding regions). At minimum the consequences of these mismatch effects should be discussed.

We thank the reviewer for raising this important subject. This question is complex, and some of the discussion is beyond the scope of the current manuscript. However, we aim to address the reviewer's question fully in this reply.

It is clear that both partial matches and intergenic matches affect an sgRNA's tendency towards depletion, especially with respect to other sgRNAs targeting the same gene. There are two distinct effects:

1. *Cumulative toxicity: A gradual increase in median depletion observed when there are many such matches, as illustrated by Kim and Hart(Kim and Hart 2021) in Fig. 2a.*

2. “Strong off-target effects”: A non-gradual, stark departure from the behavior of other sgRNAs targeting the gene, which becomes more probable as the number of such matches increases.

In practice, CERES addresses effect 1 for intergenic matches with its copy number correction, although as Fortin et al point out it does not completely remove the trend. It is unclear whether effect 1 really is a copy number effect (given the findings of Gonçalves et al. that cutting toxicity is driven only by tandem duplication, perhaps not(Gonçalves et al. 2019)). CERES does not handle the more subtle partial mismatch depletion which is also part of effect 1.

The structure of Chronos means that it does not apply an sgRNA-level copy number correction and so does not address effect 1. In response to the reviewer’s request, we tried a “pre-hoc” correction of the data, where we calculated per-sgRNA log fold change, trained a linear model fitting the difference between sgRNAs with and without partial/intergenic matches, and then constructed a pseudo-readcount matrix from the residuals. This is very similar to the approach of BAGEL2, substituting log fold changes for log Bayes factors. Although we confirmed that we could successfully detrend the data, this produced almost no effect on Chronos’s quality metrics. We believe this is because Chronos’s internal degrees of freedom already allow it sufficient flexibility to handle this cause of sgRNA disagreement. We therefore decided not to include this option in Chronos to avoid increasing complexity.

Effect 2 is more pernicious, particularly for CERES since it is prone to focus gene effect estimates on the most extreme sgRNAs rather than the consensus of sgRNAs for the gene. The probability of a given sgRNA eliciting a strong off-target effect increases as the number of partial matches increases, but the effects are highly variable and difficult to predict.

To investigate whether we could systematically identify and remove such problematic sgRNAs we trained a model to predict sgRNAs eliciting strong off-target effects. We identified off-targets in the Avana and KY data as sgRNAs that 1) are strongly depleted (log fold-change < -1) in at least 3 cell lines but 2) their target is unexpressed in those lines, and 3) additionally have median log fold-change at least 0.5 less than the median of all sgRNAs targeting that gene. There were 730 such sgRNAs in Avana and 935 in KY. Training a random forest to classify these outliers vs other sgRNAs targeting the same genes using the number of 1-, 2-, or 3-bp mismatches yields mixed results, with ROC 0.771 in Avana and just 0.617 in KY. Interestingly, the number of 3 base-pair tolerant partial matches is predictive in both libraries. This suggests that in some cases Cas9 may be tolerating more substantial mismatches than are usually considered. In the specific example we present in the revised manuscript, TCEAL7, no sgRNA had any 1-bp mismatches in hg38, but the offending sgRNA did have an unusually high number of 3-bp mismatches which may have permitted the off-target behavior of the sgRNA.

Identifying outlier sgRNAs using the number of partial matches. **Left:** SHAP values indicating per-sample feature importance estimates for the number of one-, two-, or three-base-pair mismatch-tolerant matches found in the genome, as assigned by a random forest model trained to classify sgRNAs as outliers or not. (Top: Avana; Bottom: KY) **Right:** If one used the random forest's predictions to remove sgRNAs from the libraries, how many sgRNAs would need to be removed (x-axis) to achieve a given reduction in the fraction that are outliers among remaining sgRNAs (y-axis).

Given this result, we believe it is unlikely that these outlier sgRNAs are driven by a gradual cumulative effect like cutting toxicity. We think it is appropriate to describe such sgRNAs as having off-target activity, as Fortin et al. do. It is clear that developing stronger predictors of off-target activity than simple counts of partial matches will be a significant consideration in future library designs. In the meantime it is important that CRISPR inference algorithms be robust to the possibility of strong off-target activity. We designed Chronos's sgRNA efficacy regularization with this in mind, and can report success at mitigating these effects. An example is now provided in the new Figure 6.

A much abbreviated version of this reply has been added as a paragraph to the Discussion section.

- The legend and figure for Figure 5 do not match so it is impossible to make sense of this figure.

We apologize, and have corrected this figure.

- Partially because of the issues with Figure 5 it is hard to make sense of the oncogene addiction analysis. It is also unclear what the criteria for selection of oncogenes was - in the methods it says "Finally, the 69 surviving oncogenes were manually curated to identify oncogenes known to induce oncogene addiction, leaving 40 genes". How exactly did the authors manually curate the list of 69 genes?

We removed 1) genes where the annotated alterations do not sufficiently capture the conditions where the gene becomes a dependency (for example, MDM2's own features do not adequately indicate when it is an expected dependency) 2) genes known to act as tumor suppressors in at least some contexts, 3) genes whose function is expected to be significant only in vivo, 4) genes whose function is chiefly to confer drug resistance, and 5) genes with insufficient evidence of oncogenicity in the literature. We have added this explanation to the Methods and added a table with the (now 73) genes that reached the manual curation step to the supplementary data for this manuscript. The table specifies whether they were excluded and why they were excluded.

Overall it seems that Chronos does not perform significantly better than the other approaches at identifying oncogene addiction effects.

For individual genes the comparisons are generally underpowered with 2 to 4 expected dependent lines per gene, particularly given that we expect algorithmic improvements in any given gene's essentiality profile to be relatively modest in magnitude. Nevertheless, when the data are pooled across genes, we found Chronos to be significantly better than all other approaches at identifying oncogene addiction effects in Project Achilles under all three measures; however, for the Project Score data, the differences between the first and second-best methods were not significant by any measure.

It would be interesting to see correlations between the Chronos and other metrics for individual genes to see if they are good at distinguishing the effects for the same or different genes (similar to what was done in Fig 4)

We have added two supplementary figures 3 and 4 comparing the quantitative performance between each algorithm for each expression addiction in each dataset. Overall there are no strong outliers in either direction between CERES and Chronos in either Achilles or Project Score, matching the generally high per-gene correlation between algorithms seen in Figure 6.

- The authors create a 'post-hoc' method for correcting for copy number biases (p22, Figure 6). It's not clear to me if the rest of the figures in the paper (e.g. Fig 2) make use of the Chronos scores with this post-hoc copy number correction. It's important to clarify this as potentially the copy number correction improves / worsens the quality of the results.

We apologize for the confusion. We used the copy number correction everywhere except in the

single cell line analyses (Fig. 3 and Supplementary Fig. 2) as it is not possible to infer common essentiality with only one cell line (though we note that Chronos users would be able to use estimates of common essentiality from large-scale screening datasets like Achilles and SCORE for these purposes). We have added explicit statements in the text where copy number correction has not been used.

Minor comments

- In text (main and methods) and the Fig. 1 legend, i and j are both used interchangeably to refer to a specific sgRNA.

We have changed the annotation to consistently use j

- p16 typo - 'versys'

Corrected

- p42 dated preprint reference, Goncalves et al is now published in Genome Biology

We have updated the reference

- p43 broken reference to Wu et al

We have updated the reference.

- 'Rare clonal outgrowth' is mentioned in early on (p10) but only explained in the methods, this should be explained earlier on.

We have added a sentence explaining this effect to the Introduction.

- The package on GitHub is well described and has a complete example notebook which will facilitate usability. One additional thing that would be useful to note for future users is an approximate sample runtime for Chronos, e.g. for DepMap 20Q4.

We have added this information to the README.

Minor comments on figures:

- It's not possible to visually distinguish between the CERES and Chronos box plots in Fig. 2f, perhaps outliers should not be included in this graph?

We have removed the outliers from this plot

- Fig. 2g: y-axis label should read "Recall at 90% Precision" for clarity.

We have made this change

- It would be clearer if Fig. 3 used the same color assignment for different methods as used in Fig. 2

We have changed the color scheme to be consistent.

- It's not clear why the arrows to indicate the direction of increasing performance are only used in Fig. 4, after those metrics have already been introduced in previous figures.

We have moved these arrows to Figure 2, where the metrics are first introduced.

- The box plots in Fig 6c could be scaled up a bit, they're very small compared to Fig. 6ab..

We have edited this panel to make the details easier to read.

Second round of review

Reviewer 2

The authors have addressed all of my concerns.